# Carbon Quantum Dots from Roasted Coffee Beans: Their Degree and Mechanism of Cytotoxicity and Their Rapid Removal Using a Pulsed Electric Field

**DOI:** 10.3390/foods12122353

**Published:** 2023-06-13

**Authors:** Ling Chu, Yu Zhang, Leli He, Qingwu Shen, Mingqian Tan, Yanyang Wu

**Affiliations:** 1Key Laboratory for Food Science and Biotechnology of Hunan Province, College of Food Science and Technology, Hunan Agricultural University, Changsha 410128, China; chu13257326523@163.com (L.C.); zhangyu0918@stu.hunau.edu.cn (Y.Z.); heleli@stu.hunau.edu.cn (L.H.); yaoyao3153@aliyun.com (Q.S.); 2School of Food Science and Technology, National Engineering Research Center of Seafood, Collaborative Innovation Center of Seafood Deep Processing, Dalian Polytechnic University, Dalian 116034, China; mqtan@dlpu.edu.cn; 3Horticulture and Landscape College, Hunan Agricultural University, Changsha 410128, China; 4Hunan Co-Innovation Center for Utilization of Botanical Functional Ingredients, Changsha 410128, China; 5State Key Laboratory of Subhealth Intervention Technology, Changsha 410128, China

**Keywords:** carbon quantum dots, coffee, cytotoxicity, pulsed electric field

## Abstract

Carbon quantum dots (CQDs) from heat-treated foods show toxicity, but the mechanisms of toxicity and removal of CQDs have not been elucidated. In this study, CQDs were purified from roasted coffee beans through a process of concentration, dialysis and lyophilization. The physical properties of CQDs, the degree and mechanism of toxicity and the removal method were studied. Our results showed that the size of CQDs roasted for 5 min, 10 min and 20 min were about 5.69 ± 1.10 nm, 2.44 ± 1.08 nm and 1.58 ± 0.48 nm, respectively. The rate of apoptosis increased with increasing roasting time and concentration of CQDs. The longer the roasting time of coffee beans, the greater the toxicity of CQDs. However, the caspase inhibitor Z-VAD-FMK was not able to inhibit CQDs-induced apoptosis. Moreover, CQDs affected the pH value of lysosomes, causing the accumulation of RIPK1 and RIPK3 in lysosomes. Treatment of coffee beans with a pulsed electric field (PEF) significantly reduced the yield of CQDs. This indicates that CQDs induced lysosomal-dependent cell death and increased the rate of cell death through necroptosis. PEF is an effective way to remove CQDs from roasted coffee beans.

## 1. Introduction

Coffee is a beverage made from roasted coffee beans. Today, coffee is grown in more than 70 countries and is very popular around the world. Moderate coffee consumption has been found to protect against cardiovascular disease [1], metabolic syndrome and type 2 diabetes [2], and has antioxidant and anti-inflammatory effects. However, it also contains a small amount of hazards, such as acrylamide, 2,3-butanedione and carbon quantum dots (CQDs). It has been reported that dietary acrylamide increases the risk of ovarian, kidney and lung cancers, as well as cancers related to the digestive and respiratory systems [3]. Chronic exposure to high concentrations of 2,3-butanedione, a hazard in roasted coffee, increases the risk of obstructive bronchiolitis for coffee workers [4]. Coffee also contains CQDs, which have a known toxicity.

CQDs are nanomaterials with a diameter less than 10 nm. They have the advantages of excellent optical properties, low cost and good biocompatibility, which make them potentially valuable in biosensing, bioimaging, photocatalysis and drug delivery [5,6,7]. However, CQDs also cause cytotoxicity. CQDs with different sizes, shapes and physical and chemical properties have been found in baked goods such as hamburgers [8], roast beef [9], roast lamb [10], pizza, instant coffee [11] and caramel. The higher the heating temperature is, the smaller the size of the CQDs and the greater the toxicity. CQDs accumulated in the kidney, lung, brain, liver and small intestine of mice that were administered CQDs [12]. Toxicity and cytotoxic mechanisms can be determined using cellular models. To date, many types of cell death have been discovered [13]. Programmed cell death is divided into caspase-dependent cell death (such as pyroptosis and apoptosis) and caspase-independent cell death (such as necroptosis, autophagy-dependent cell death, oncosis, ferroptosis, lysosomal-dependent cell death, parthanatos, oxeiptosis and alkaliptosis). Some nanomaterials have been found to enter cells and cause toxicity in mice or cells, such as gold nanoparticles [14], nanocopper [15], nanocellulose [16] and SiO_2_ nanoparticles [17]. Studies have shown that LLOMe and silica nanoparticles can enter cells and cause lysosomal rupture, which leads to inflammation, oxidative stress and apoptosis or necrosis [18]. Treatment of macrophages with carbon black nanoparticles results in lysosomal rupture, cathepsin B release, reactive oxygen species production and reduced intracellular ATP levels [19]. CQDs from various raw materials, such as roasted salmon, lamb, chicken, fish and Penaeus vannamei, can also enter cells and produce certain toxicity in cells [20,21]. However, the degree and mechanism of cytotoxicity of CQDs extracted from coffee beans remain to be elucidated.

Pulsed electric field (PEF) technology, a new nonhot food processing technology, can be used to process foods at a low temperature and has little influence on the colour, aroma, taste and nutritional composition of food [22]. It has high efficiency, low energy consumption, and fewer byproducts, and it contributes no pollution to the environment while fully maintaining the flavour characteristics of foods. It can be applied to almost any temperature-sensitive food, such as fruits and vegetables [23]. At present, PEF technology can be applied in the fields of food preservation, assisted fermentation, assisted extraction, assisted drying, assisted freezing, thawing and nonthermal sterilization. Studies have shown that PEF can passivate enzyme activity through destroying the secondary structure and dipole moment of certain bonds in the enzyme. Moreover, there are electrochemical problems and electrode corrosion problems in the process of PEF treatment of food, a series of oxidation reactions and a series of complex oxidation products, which have an impact on food lipid components [24].

There are many ways to roast coffee beans, including American, Viennese, French, Italian, and Spanish roasting methods. Coffee beans can be divided into Arabica beans and Robusta beans. The aim of this study was to evaluate the degree, mechanism of toxicity of CQDs and the method for removing CQDs from roasted coffee beans. Our findings suggested that the longer roasting time of coffee beans, the smaller the size of CQDs and the greater the toxicity. CQDs induced lysosomal-dependent cell death and increased the rate of cell death through necroptosis. The CQDs can be removed via the PEF method, which may provide theoretical guidance for coffee production and lay a theoretical foundation for preventing the harm of endogenous nanoparticles in food to human health.

## 2. Materials and Methods

### 2.1. Reagents

Coffee beans were bought from Yunli Coffee Company (Yunnan, China). Z-VAD-FMK (S7023), RIP (D94C12) XP (3493S) and RIP3 (D8J3L) (15828S) have been reported in previous literature [20]. The Lysosensor Green DND-189 probe was purchased from Thermo Fisher Scientific Co. Ltd. (Waltham, MA, USA). Dulbecco’s modified Eagle’s medium (DMEM) and fetal bovine serum (FBS) were obtained from Biological Industries (BioInd; Kibutz Beit Haemek, Israel). Anti-GAPDH antibody (ZB002) and propidium iodide (PI)-Annexin V/fluorescein isothiocyanate (FITC) apoptosis detection kit (556547) were obtained from Shanghai Beyotime Biotechnology Co., Ltd. (Shanghai, China). Goat anti-mouse IgG (1036-05) and goat anti-rabbit (4050-05) were obtained from Southern Biotechnology Associates, Inc. (Birmingham, AL, USA). DQ^TM^ red bovine serum albumin (BSA) (D12051) was purchased from Thermo Fisher Scientific Co., Ltd. (Waltham, MA, USA).

### 2.2. Instruments

TEM (JEM-2100F) was purchased from Japan Electronics (JEOL) Co., Ltd. (Tokyo, Japan). Ultraviolet spectrophotometer (UV-2600) was purchased from Shimadzu Instruments (Suzhou) Co., Ltd. (Suzhou, China). Fluorescence spectrophotometer (F-7000) was obtained from Hitachi, Ltd. (Tokyo, Japan). X-ray diffractometer (XRD-6000) was purchased from Shimadzu Corporation (Kyoto, Japan). X-ray photoelectron spectrometer (Nexsa) was purchased from Thermo Fisher Scientific Co., Ltd. (Waltham, MA, USA). Flow cytometer (Beckman MoFlo XDP) was purchased from Carl Zeiss AG (Oberkochen, Germany). Pulsed electric field equipment was obtained from the School of Food Science and Engineering, SCUT, China.

### 2.3. Preparation of Coffee Beans Samples

The coffee beans belong to the Arabica species and is cultivated in Yunnan province in China. The raw coffee beans were roasted in the oven at 210 °C for 5, 10 and 20 min, respectively, and then grinded with a coffee beans grinder.

### 2.4. The Extraction and Purification of CQDs

The CQDs were obtained as our previous reported [12]. Briefly, appropriate amount of coffee beans powder was soaked in anhydrous ethanol for 24 h, followed by filtration and rotary evaporation. Then, the crude CQDs were extracted with ethyl acetate. After centrifugatiton at 8000× *g* rpm for 10 min, the supernatant was collected and macroporous resin column was used to adsorb impurities. Additionally, the collected solution was concentrated through rotary evaporation, and then dialysis bag was used. Finally, the CQDs were obtained after freeze-dried to obtain and stored at −20 °C.

### 2.5. Characterization of Structural Properties of CQDs

The CQDs was added to the ultra-thin carbon film, and the morphology of CQDs was observed via transmission electron microscope (TEM) after natural drying. Then, the size of CQDs were analyzed using Image J. The CQDs and potassium bromide (KBr) were mixed and dried at a ratio of 1:100, and then scanned in the range of 500–4000 cm^−1^ to obtain the infrared spectrum of CQDs. The CQDs were filled into the X-ray diffractometer (XRD) glass sample tank with a depth of 0.5 mm, and the surface was compacted and scraped flat with a glass sheet. The radiation source of the XRD is CuKα. The wavelength of the X-ray is 0.154 nm, and the 2θ diffraction angle ranges from 10° to 80°.

### 2.6. Testing the Apoptosis of NRK Cells

The NRK cells were seeded in 12-well culture plates and incubated with CQDs for 12 h. The cells in culture medium and about 1 × 10^6^ attached cells were collected into the appropriate centrifuge tube and washed 3 times with PBS. Then, the cells were stained with 5 μL PI and annexin V-FITC for 15 min. The total fluorescence intensity in the cells was tested using flow cytometry.

### 2.7. Testing the Rates of Cell Death

The NRK cells were seeded in 12-well culture plates and about 1 × 10^6^ attached cells were collected into the appropriate centrifuge tube and washed with PBS 3 times. Then, the cells were stained with 5 μL PI. The total fluorescence intensity in cells was tested using flow cytometry.

### 2.8. Testing of Lysosomal Enzyme Activity

About 1 × 10^6^ NRK cells were seeded in glass chamber and treated with CQDs for 12 h and stained with DQ-Red BSA (10 µg/mL) for 3 h. The red fluorescence intensity in the cells was observed using confocal microscopy. The fluorescence intensity of each image was statistically analysed with Image Pro-Plus 6.0.

### 2.9. Testing of Lysosomal pH Value

The NRK cells were seeded in 12-well culture plates and treated with CQDs for 12 h and stained with the lysosomal pH probe 1 µM lysosensor green DND-189 for 30 min. The total green fluorescence intensity in the cells was measured using flow cytometry.

### 2.10. PEF Treatment for CQDs

The raw coffee beans were roasted in the oven at 210 °C for 20 min, and grinded with a coffee beans grinder. Then, the powder were divided into 4 groups with 100 g in each group and put it into the sample dish of PEF with a sample height of 10 mm. Then, each sample was treated with PEF for 50 times after added with 100 mL distilled water with the corresponding voltage 110 V, 150 V or 210 V. The ball distance of each group was 1.2, 1.8 or 2.4 mm. Finally, the CQDs were purified and obtained with the method described above. Each experiment was repeated at least 3 times.

### 2.11. Western Blotting

NRK cells were seeded in 6-well culture plates and incubated with CQDs for 12 h. The medium containing CQDs was removed, and the cells were washed with PBS. Western blotting was carried out according to a previously reported method [20]. Briefly, the membranes were incubated with 5% skim milk powder for 1 h and incubated with primary antibody for 1 h. Then, the membranes were incubated with secondary antibody (Goat anti-mouse IgG or goat anti-rabbit antibody) for 1 h after washing with TTBS three times. Then, the membranes were imaged and analysed using an ultrasensitive chemiluminescence instrument (Sweden, Model: Serial No. 3614294).

### 2.12. Statistical Analysis

All data were analysed using the software of IBM SPSS Statistics 23 with one-way analysis of variance (ANOVA). Error bars, S.D. Differences with *p* < 0.05 represented statistical significance. Each experiment was repeated at least 3 times.

## 3. Results

### 3.1. Characterization of CQDs from Roasted Coffee Beans

CQDs were extracted and purified from roasted coffee beans. To investigate the morphology of CQDs, we measured the diameter of the carbon dots using transmission electron microscopy (TEM). The results showed that CQDs from coffee beans roasted at 210 °C for 5 min, 10 min or 20 min were nearly spherical and evenly dispersed without obvious aggregation (Figure 1A–C). The average sizes of CQDs roasted for 5, 10 and 20 min were 5.69 ± 1.10 nm, 2.44 ± 1.08 nm and 1.58 ± 0.48 nm, respectively. The size of CQDs from coffee beans roasted at 210 °C for 10 min was inconsistent (Figure 1D–F). This showed that at the given baking temperature, the size of the CQDs decreased, and the shape was more stable with a longer baking time.

To explore the physical and optical properties of CQDs, they were measured via ultraviolet spectrophotometry and fluorescence spectrophotometry. The results showed that when the excitation wavelength moves from 360 nm to 430 nm, the redshift of the CQD emission peak appears. When the emission wavelength is 450 nm, the excitation wavelength is 370 nm and CQDs show the highest emission intensity (Figure 2A–C). This is a common phenomenon observed in CQDs due to inhomogeneous size distribution or different surface defect states of the particles resulting in differences in electron transition paths. The UV-vis absorption value of CQDs was approximately 300 nm, and the CQDs roasted 210 °C for 5 min retained the highest photoluminescence characteristics (Figure 2D).

The CQDs were also tested using Fourier transform infrared spectroscopy (FTIR) and X-ray photoelectron spectroscopy (XPS) to explore the group compositions and elemental composition of CQDs, respectively. The results showed that the -OH group appeared at 3376 cm^−1^. The absorption peaks at 2925 cm^−1^ and 2857 cm^−1^ were attributed to the -CH_2_ group. The absorption peaks at 1734 cm^−1^, 1636 cm^−1^ and 1411 cm^−1^ were attributed to C=O, C=C and -CH_3_, respectively. The absorption peak at 1055 cm^−1^ was assigned to C-O-C. The intensity of peaks for O-H, CH_2_ and C=O at 20 min declined compared with those of the peaks for O-H, CH_2_ and C=O at 5 min or 10 min, which might be due to the increased degree of carbonization and partial breakdown of C-O-C and N-H. (Figure 3A). X-ray diffraction (XRD) of CQDs at 5, 10 and 20 min showed that the peak values were 20°, 20° or 22°, respectively, indicating that CQDs were in an amorphous state (Figure 3B).

X-ray photoelectron spectroscopy (XPS) analysis showed that three peaks, C_1s_, N_1s_ and O_1s_, were observed at 283.08, 402.08 and 533.08 eV for the CQDs roasted for 5 min, and the element composition was 78.03%, 0.85% and 26.07%, respectively (Figure 4A and Table 1). Three peaks, C_1s_, N_1s_ and O_1s_, were observed at 285.08, 399.08 and 531.08 eV for the CQDs roasted for 10 min, and the element composition was 61.23%, 0.44% and 38.34%, respectively (Figure 4B). Three peaks, C_1s_, N_1s_ and O_1s_, were observed at 285.08, 400.08 and 533.08 eV for the CQDs roasted for 20 min, and the element composition was 79.58%, 1.48% and 18.94%, respectively (Figure 4C). High-resolution mass spectrometry analysis of C_1s_ showed the presence of C=C, C-N, C-O and C=O bonds at binding energies of 284.8, 285.5, 286.4 and 288.5 eV, respectively (Figure 4D–F). High-resolution analysis of N_1s_ showed the presence of pyridinic N with a binding energy of 397.9 eV, amino N at 399.3 eV and pyrrolic N at 400.4 eV (Figure 4G–I). High-resolution analysis of O_1s_ showed the presence of *O-C=O, O-C=O* and C-O bonds with binding energies at 531.4, 532.3 and 533.2 eV, respectively (Figure 4J–L).

### 3.2. CQDs Increased the Rate of Apoptosis and Cell Death

After oral administration of CQDs in mice, they mainly accumulate in organs such as the brain, kidney, liver and small intestine. Thus, NRK cells, normal epithelial cells from the kidney, were chosen as the cell model to elucidate the toxicity and toxic mechanism of CQDs. To assess the toxicity of CQDs, NRK cells were treated with CQDs for 12 h. The apoptosis rate of the control group was 4.28%. When the roasting time of CQDs was 5 min, the cell apoptosis rates were 6.14%, 4.83% and 6.81% at 0.1, 0.5 and 1.0 mg/mL CQDs, respectively, which were not significantly different compared to the control group. The apoptosis rates of cells increased to 4.93%, 7.39% and 10.11% at the same concentration of CQDs, respectively, in the CQDs roasted for 10 min. However, the apoptosis rates of the cells increased to 8.75%, 10.61% and 12.28% after the cells were treated with CQDs for 20 min, respectively, which were significantly different from those of the control group (Figure 5A,C). The rate of apoptosis increased with increasing roasting time and concentration of CQDs.

The rate of cell viability also reflects the toxicity of CQDs. The cell viability rate of the control group was 94.74%. When the roasting time of CQDs was 5 min, the cell viability rates were 92.92%, 93.10% and 91.26% within the 0.1, 0.5 and 1.0 mg/mL CQD-treated groups, respectively. The rates of cell viability were reduced to 90.68%, 88.30% and 87.30% at the same concentration of CQDs, respectively, when the CQDs were roasted for 10 min, which were significantly different compared to the control group and 5 min group. However, when the roasting time of CQDs was 20 min, the cell viability rates were reduced to 92.92%, 93.10% and 91.26% in 0.1, 0.5 and 1.0 mg/mL CQD-treated groups, respectively, which were significantly different from those of the other groups (Figure 5A,D). So, we concluded that the rate of cell viability also increased with increasing roasting time and concentration of CQDs.

The effect of CQDs on cell death was analysed via flow cytometry with a PI probe. The cell death rate of the control group was 3.17%. When the roasting time of CQDs was 5 min, the cell death rates were 4.48% and 4.67% at 0.1 and 0.5 mg/mL CQDs, respectively, which were not significantly different compared to the control group. However, when the concentration of CQDs was 1 mg/mL, the cell death rate was 6.36%, which was significantly different from the control group. The rates of cell death increased to 6.59%, 7.93% and 6.92% at 0.1, 0.5 and 1.0 mg/mL CQDs, respectively, when CQDs roasted for 10 min. However, the rates of apoptosis increased to 8.20%, 9.65% and 11.41%, respectively, after the cells were treated with CQDs roasted for 20 min, which were significantly different from the control group. The rate of cell death also increased with increasing roasting time and concentration of CQDs (Figure 5B,E). These data implied that the CQDs roasted for 5 min caused lowest cytotoxicity. It also indicated that the longer the roasting time and the higher the concentration of CQDs, the greater the cytotoxicity. CQDs roasted for 20 min had significantly higher cytotoxicity.

### 3.3. CQDs Induced Caspase-Independent Cell Death

Programmed cell death is divided into caspase-dependent cell death and caspase-independent cell death [25]. Phosphatidyl serine is mainly distributed on the inner side of the cell membrane, near the cytoplasm. Apoptosis is one of the ways of cell death. In the early stage of apoptosis, phosphatidylserine is reversed outside the cell membrane, and annexin V can bind to inverted phosphatidylserine. A green fluorescent labelled annexin V-FITC probe, flow cytometry and fluorescence microscopy are used to test whether phosphatidylserine was everted. In the late stage of cell apoptosis, the cell membrane is damaged, and a propidium iodide (PI) probe can penetrate the cell membrane and bind to chromosomes in the nucleus. Necrosis is a passive cell death under environmental stress. Necrosis can also be regulated via fine cellular signalling pathways. Necrosis is caused by the damage of cell membrane, and then the cell permeability is changed. The PI probe can pass through the cell membrane and combine with the chromosome in the nucleus, and the cell is labelled with PI. So, the rate of cell death may include the rates of apoptosis and necrosis. Z-VAD-FMK is an inhibitor of irreversible caspases and can penetrate cell membranes and inhibit apoptosis caused by the activation of caspase [26]. To study the effect of Z-VAD-FMK on the rate of cell apoptosis, NRK cells were treated with CQDs or CQDs plus Z-VAD-FMK, and the apoptosis rate was analysed via flow cytometry. The results showed that CQDs promoted rates of apoptosis from 2.40% to 8.46%, and the cell death rate also increased from 2.33% to 10.83%, but there was no difference between the CQD-treated group and CQDs plus Z-VAD-FMK-treated group (Figure 6A–D). Therefore, we concluded that CQDs from roasted coffee beans induced caspase-independent cell death.

### 3.4. CQDs Induced Lysosomal-Dependent Cell Death

To test the effect of CQDs on lysosomes, NRK cells were treated with CQDs extracted from coffee beans, and lysosomal function was tested. DQ-red BSA is a fluorescent probe for detecting lysosomal activity. When it enters the lysosome, it is cleaved by proteases within the lysosome, resulting in the release of fluorescent fragments that emit bright fluorescence. The higher the lysosomal enzyme activity is, the stronger the fluorescence intensity. The results showed that the red fluorescence intensity in the CQDs-treated groups was weaker than that of the control, indicating that CQDs decreased lysosomal activity (Figure 7A,B).

Lysosensor Green DND-189 is a green fluorescent probe that fluoresces in acidic compartments and accumulates in acidic organelles. The lower the pH value is, the stronger the green fluorescence. To test the effect of CQDs on lysosomal pH, cells were treated with CQDs and stained with 1 μM Lysosensor Green DND-189 for 30 min. The green fluorescence intensity of the cells was measured via flow cytometry. The fluorescence intensity of the CQD-treated group was weaker than that of the control group. The fluorescence intensity in CQDs from 5 min roasting was higher than that of CQDs from 10 or 20 min roasting. Our data indicated that CQDs increased the pH value of lysosomes. According to these data, we concluded that the CQDs destroyed lysosome function (Figure 7C).

Programmed necrosis is regulated by mixed-lineage kinase domain-like protein (MLKL) and receptor-interacting protein kinase (RIPK). Tumour necrosis factor-α (TNF-α) binds with tumour factor receptor (TN-FR) to lyse RIPK1 and promote phosphorylation of RIPK3. Then, it combines with MLKL to form the RIPK1-RIPK3-MLKL complex, namely, the necrosome. Necrosomes migrate from the cytoplasm to the cell membrane or organelle membrane and trigger a series of downstream actions, including destruction of membrane integrity and cytoplasmic ATP degradation [27]. Studies have reported that lysosomal dysfunction can cause the accumulation of RIPK1, RIPK3 and MLKL in lysosomes, leading to programmed cell necrosis [28]. To explore whether CQDs could cause RIPK1 and RIPK3 accumulation in lysosomes, we measured the protein expression of RIPK1 and RIPK3 via Western blotting after the cells were treated with CQDs. Our results showed that the protein expression in the CQD-treated groups was higher than that in the control group. The protein expression in CQDs roasted for 20 min was higher than that of CQDs roasted for 10 min or 5 min (Figure 7D,E). These results indicated that CQDs could inhibit the degradation of RIPK1 and RIPK3 and lead to accumulation of these proteins in lysosomes.

### 3.5. PEF as an Effective Way to Remove CQDs

PEF is a new nonthermal food processing technology. It can be used for inactivation of vegetative microbial cells, inhibition of bacterial attachment to stainless steel plates, extraction of bioactive compounds, targeting of Staphylococcus aureus inoculated in milk, ablation of hepatocellular carcinoma and as an alternative mild preservation technology. Here, we report that the CQD content in the PEF-treated group was lower than that in the control group (Figure 8). When the electric field intensity was 3.0 kV/cm, the yield of CQDs was lowest. Our results showed that the electric field intensity can reduce the yield of CQDs. Therefore, we speculate that PEF is an effective technology to remove CQDs.

## 4. Discussion

Both food thermal processing and artificial synthesis can produce CQDs. The synthesis method of CQDs includes “top-down” and “bottom-up”. The carbon sources of CQDs synthesized in the “top-down” method are generally graphite rods, carbon fibres, activated carbon, carbon nanotubes, etc. These carbon-rich substances are decomposed through laser erosion and arc discharge. “Bottom-up” is the synthesis method of CQDs obtained from carbon-rich material. “Bottom-up” synthesis methods include chemical oxidation, hydrothermal/solvothermal, combustion, the template method, microwave synthesis, etc. [29]. Thermally processed food contains CQDs. In the process of thermal processing, the synthesis of carbon dots via the decomposition of large molecules and the polymerization of small molecules is similar to the “bottom-up” synthesis of carbon dots [30]. It also has been reported that CQDs can be prepared from natural resources [6,7,31,32,33,34].

The formation of CQDs is a complex process involving complex interactions between food components and the internal environment. Here, we reported that the roasting coffee beans contained CQDs. It has been reported that normal cooking methods, such as baking, roasting and grilling, may decompose food components, thus producing CQDs [9]. Food ingredients can be used as reactants for producing CQDs, including glucose, citric acid, amino acids, polysaccharides, proteins, etc. [35]. Once energy is input, such as through irradiation or heating, it drives the interactions between food components and leads to the carbonization and condensation of CQDs. The food ingredients determine the properties of CQDs in the food items. Coffee beans are rich in alkaloids and amino acids, polysaccharides and a small amount of oligosaccharides, chlorogenic acid and small molecular organic acids, coffee oil, sterols and diterpenoids. However, which components provide raw materials for the formation of carbon spots in roasting coffee beans needs to be studied. It has been reported that food components interact with each other and form fluorescent particles at the early stage. With the extension of the roasting time, carbon cores occur owing to the increased condensation and then form CQDs [36]. Some reported that the longer the cooking time, the smaller the final CQD particle size [37]. In this paper, we also found that at the given baking temperature, the size of the CQDs decreased, and the shape was more stable with a longer baking time. The formation mechanism of CQDs in roasting coffee beans remains to be elucidated.

In this paper, we also report that the longer the roasting time and the higher the concentration of CQDs, the greater the cytotoxicity. With the development of nanotechnology, the toxicity of nanoparticles has attracted more and more attention. Nanoparticles, including carbon dots, are recognized by receptors on the cell membrane after binding with proteins in serum, and enter the cell through endocytosis [38]. It is reported that liver, kidney and lung are the main target organs of nanoparticles [12,39]. They are not easily eliminated and may accumulate in the human body for a long time. Its toxicity is related to its shape, size, surface functional groups and dissolution rate [39]. Lysosomes are acidic organelles that break down material entering the cell from the outside and digest local cytoplasmic or damaged organelles within cells to maintain cell homeostasis. Extracellular substances such as pathogens can be transported to lysosomes for degradation through endocytosis, while intracellular cytosolic macromolecules or organelles are transported to lysosomes for degradation through the autophagy pathway [40]. Drugs or nanoparticles make the lysosomal membrane unstable, resulting in loss of integrity and lysosomal membrane permeability. The release of lysosomal cathepsins and hydrolytic enzymes causes lysosomal damage [41]. As a result, all contents in lysosomes are released and result in lysosomal rupture and induce lysosome-dependent cell death. Then, apoptosis, programmed necrosis, and ferroptosis can be triggered so that the cell death signal is amplified [25]. Necroptosis is regulated by RIPK1, RIPK3 and MLKL; can cleave RIPK1; promotes phosphorylation of RIPK3 and forms the RIPK1-RIPK3-MLKL complex, namely, necrosomes. CHIP regulates necroptosis through ubiquitination- and lysosome-dependent RIPK1 and RIPK3, which reduces the stability of RIPK1-RIPK3-MLKL necrosomes and inhibits the occurrence of necrotizing apoptosis [28]. Here, we report that CQDs increased the rate of apoptosis. However, the caspase-specific inhibitor Z-VAD-FMK was not able to block CQD-induced apoptosis and cell death. Further results showed that CQDs decreased lysosomal activity, increased the pH value in lysosomes and increased the protein expression of RIPK1 and RIPK3, the key regulators of necroptosis. Therefore, we concluded that CQDs in roasted coffee beans induced lysosomal-dependent cell death and expanded the rate of cell death through necroptosis.

It has been reported that PEF generates a transmembrane potential difference through applying an external electric field. After the transmembrane potential difference is formed, the membrane forms a hydrophilic pore with an unstable structure [42]. In the process of continuous electric field treatment, the pores gradually become larger, the number of pores gradually increases, and then intracellular substances leak out of the cell through the pores [22]. PEF can achieve sterilization through the interaction of an external electric field and the microbial cell membrane. PEF can disassemble α-helices and β-folding. It can decompose chromosomal DNA, linear DNA, degraded RNA and smaller molecular particles [43]. Our results also showed that PEF can reduce the content of CQDs in roasted coffee beans, although the mechanism of PEF removal of CQDs from roasted coffee beans remained to be explored.

## 5. Conclusions

In summary, we elucidated the degree and mechanism of toxicity of CQDs extracted from coffee. It was also found that the longer the roasting time was, the smaller the size and the greater the toxicity of CQDs. We confirmed that CQDs induced lysosomal-dependent cell death and increased the rate of cell death through necroptosis. As a new food processing technology, PEF can reduce CQDs from roasted coffee beans, which provides a new strategy for removing CQDs from processed foods. However, the formation mechanism of CQDs in roasting coffee beans and the mechanism of PEF removal of CQDs from roasted coffee beans remain to be elucidated.

## Figures and Tables

**Figure 1 foods-12-02353-f001:**
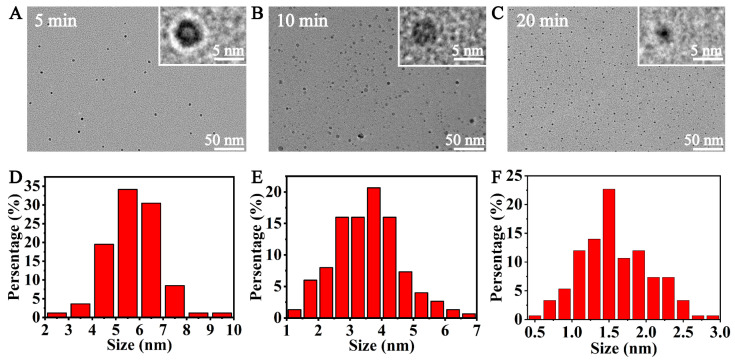
TEM analysis of CQDs. TEM image of CQDs from coffee beans roasted at 210 °C for 5 min (**A**), 10 min (**B**) or 20 min (**C**). (**D**–**F**) The size distribution of CQDs in (**A**), (**B**) or (**C**), which were analysed using ImageJ software.

**Figure 2 foods-12-02353-f002:**
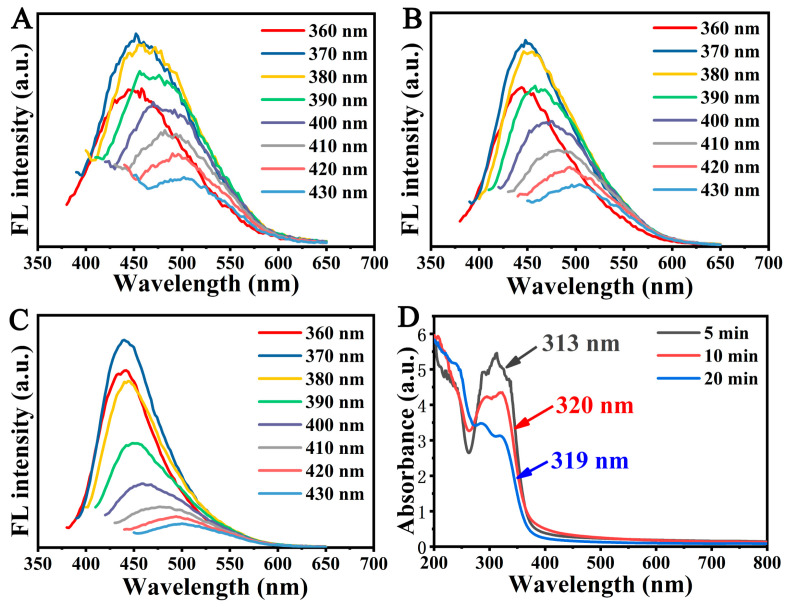
Ultraviolet and fluorescence spectra of CQDs. Fluorescence emission spectra of CQDs roasted for 5 min (**A**), 10 min (**B**) or 20 min (**C**) in the wavelength range of 360–430 nm. (**D**) The UV-vis absorption spectrum of CQDs roasted for 5, 10 or 20 min.

**Figure 3 foods-12-02353-f003:**
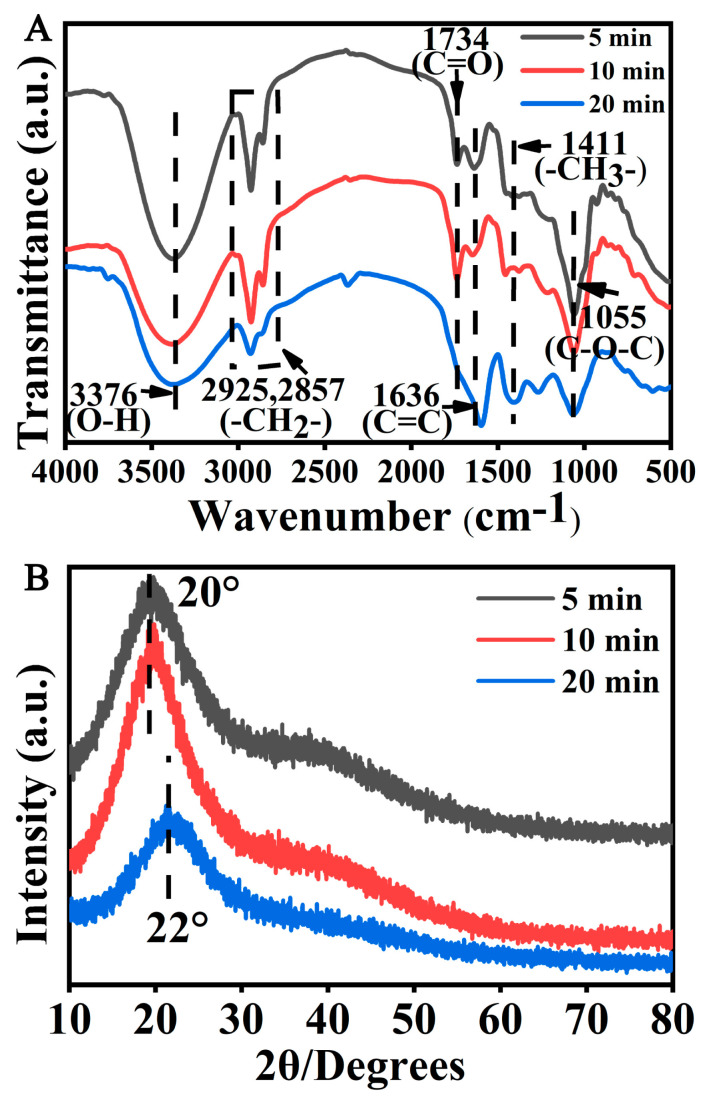
FTIR and XRD analysis of CQDs. (**A**) FTIR analysis of the CQDs from coffee beans roasted for 5, 10 or 20 min. (**B**) XRD analysis of the CQDs from coffee beans roasted for 5, 10 or 20 min.

**Figure 4 foods-12-02353-f004:**
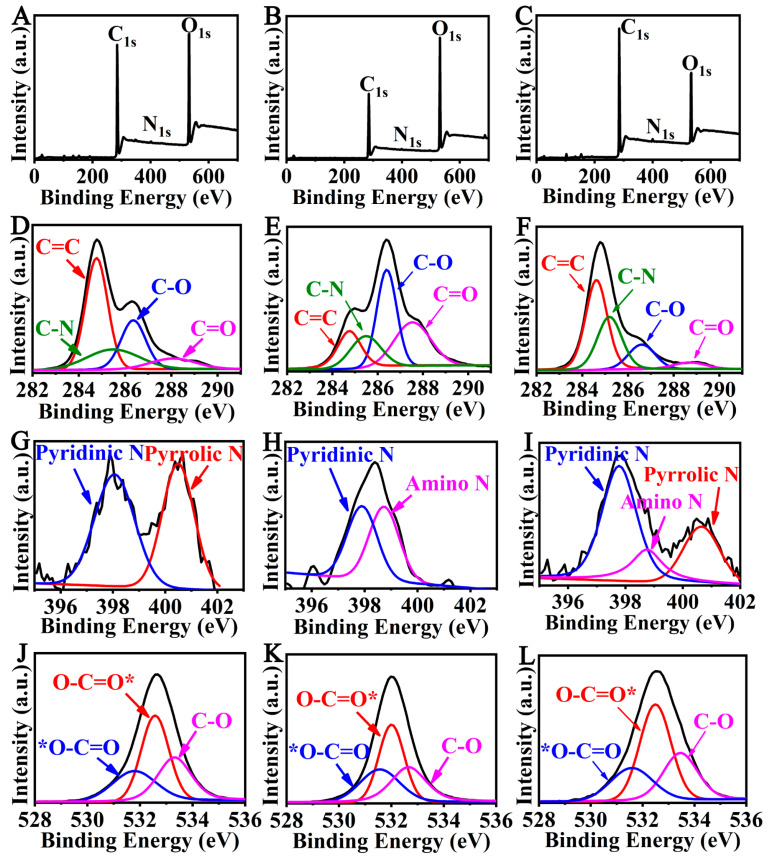
XPS analysis of CQDs. XPS spectrum of CQDs extracted from coffee beans roasted at 210 °C for 5 min (**A**), 10 min (**B**) and 20 min (**C**). (**D**–**F**) High-resolution analysis of C_1s_ of CQDs roasted for 5, 10 and 20 min. (**G**–**I**) High-resolution analysis of N_1s_ of CQDs roasted for 5, 10 and 20 min. (**J**–**L**) High-resolution analysis of O_1s_ of CQDs roasted for 5, 10 and 20 min.

**Figure 5 foods-12-02353-f005:**
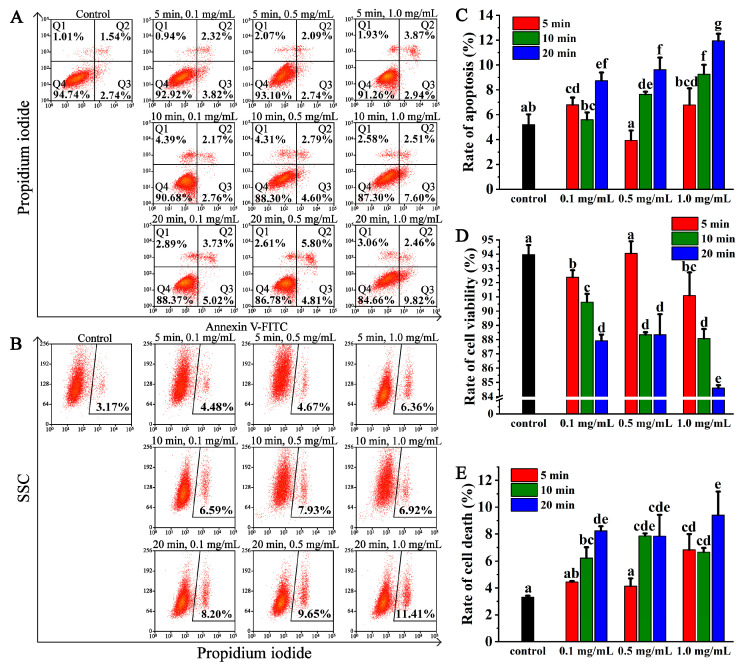
Cytotoxicity analysis of CQDs. (**A**) NRK cells were treated for 12 h with or without 0.1, 0.5 or 1 mg/mL CQDs roasted for 5, 10 or 20 min. Then, the cells were stained with PI and annexin V-FITC for 15 min, and flow cytometry was used to detect the rates of apoptosis. (**B**) NRK cells were treated as in A. The cells were stained with PI for 15 min, and the rates of cell death were analysed via flow cytometry. (**C**) The rates of apoptosis in A were analysed using IBM SPSS Statistics 25 software. (**D**) The rates of cell viability in A were analysed using IBM SPSS Statistics 25 software. (**E**) The rates of cell death in B were analysed using IBM SPSS Statistics 25 software. Different letters indicate significant differences (*p* < 0.05). Each group was repeated at least three times. Error bars, s.d.

**Figure 6 foods-12-02353-f006:**
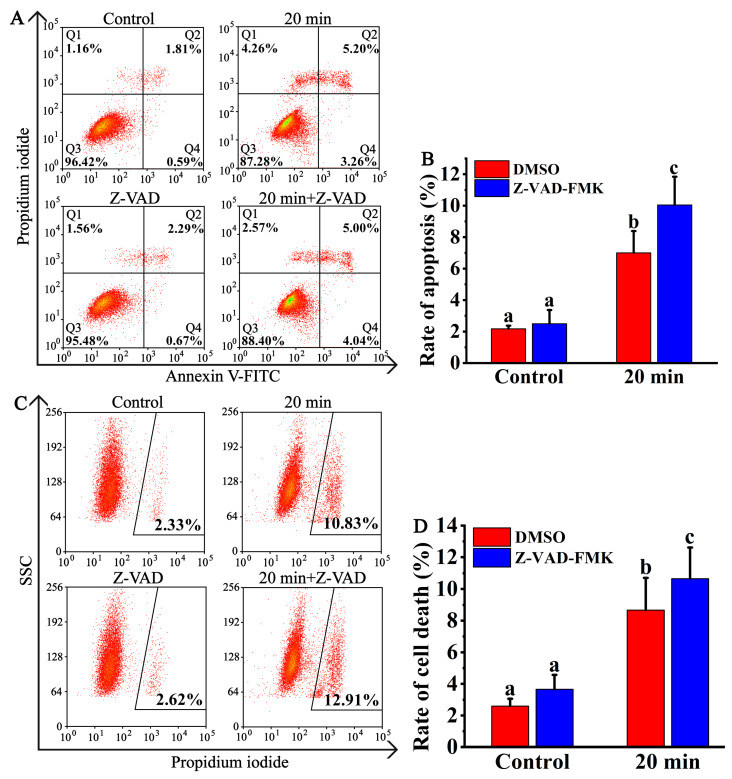
Z-VAD-FMK does not decrease CQDs-induced cell death. (**A**) NRK cells were treated with 1 mg/mL 20 min-CQDs, 10 μM Z-VAD-FMK or 1 mg/mL CQDs plus 10 μM Z-VAD-FMK for 12 h and then stained with PI and annexin V-FITC for 15 min. The rates of apoptosis were detected via flow cytometry. (**B**) The rates of apoptosis in A were analysed using IBM SPSS Statistics 25 software. Different letters indicated significant differences (*p* < 0.05). Each group was repeated at least three times. Error bars, s.d. (**C**) NRK cells were treated as in A. Then, the cells were stained with PI, and the rates of cell death were analysed using flow cytometry. (**D**) The rates of cell death in (**C**) were analysed using IBM SPSS Statistics 25 software. Different letters indicate significant differences (*p* < 0.05). Each group was repeated at least three times. Error bars, s.d.

**Figure 7 foods-12-02353-f007:**
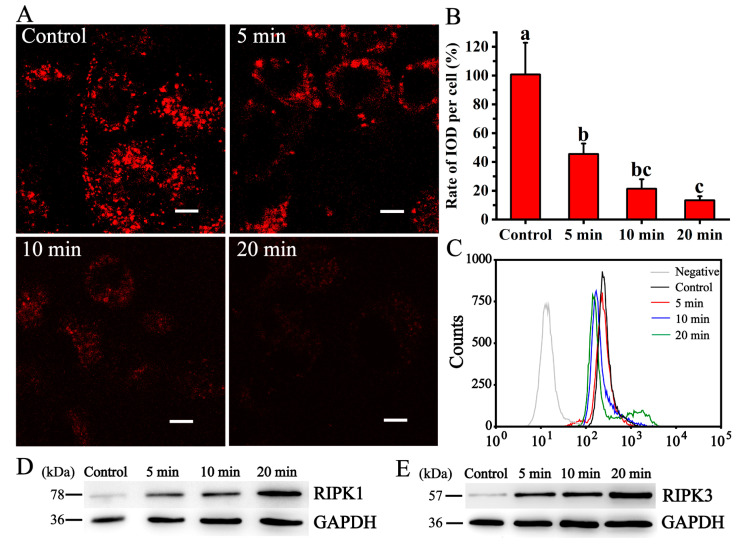
The effect of CQDs on lysosomes. (**A**) DQ-Red BSA (10 μg/mL) was used to incubate NRK cells for 3 h after the cells were treated with CQDs for 12 h. Images were obtained using an SP8 Leica confocal microscope. The scale bar is 5 μm. (**B**) NRK cells were treated as in A, and the fluorescence intensity was analysed using Image Pro-Plus 6.0. Different letters indicate significant differences (*p* < 0.05), and each group was repeated at least three times. Error bars, s.d. (**C**) NRK cells were treated with CQDs for 12 h and incubated with the lysosomal pH probe Lysosensor green DND-189 (1 μM) for 30 min. The fluorescence intensity was analysed via flow cytometry. (**D**) NRK cells were treated with CQDs for 12 h. The protein expression of RIPK1 was tested using Western blotting. (**E**) NRK cells were treated with CQDs for 12 h. The protein expression of RIPK3 was tested via Western blotting. GAPDH was used as the control.

**Figure 8 foods-12-02353-f008:**
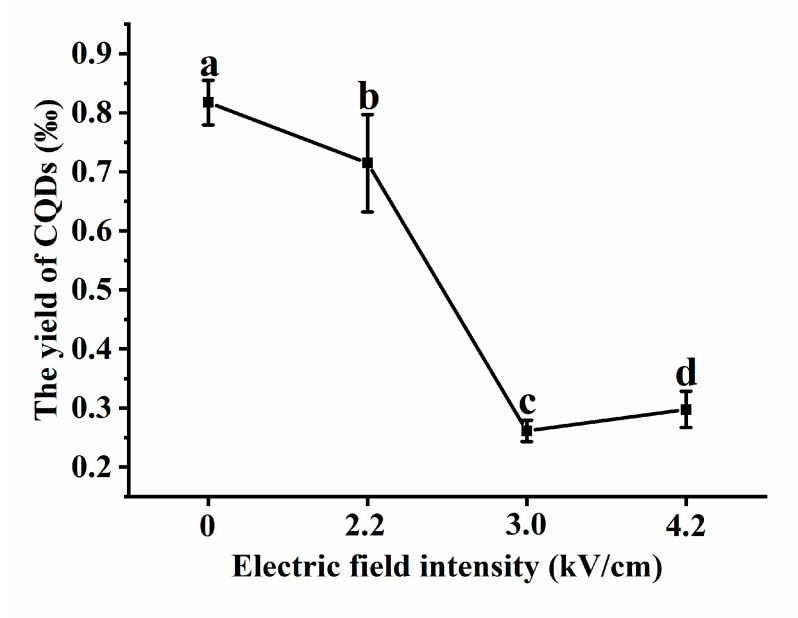
The effect of PEF on CQD yield in coffee powder. The coffee powder was treated with 2.2, 3.0 or 4.2 kV/cm PEF, and then CQDs were extracted to obtain the yield of CQDs. Different letters indicate significant differences (*p* < 0.05), and each group was repeated at least three times. Error bars, s.d.

**Table 1 foods-12-02353-t001:** Element Content of CQDs.

	The Percentage of Carbon Element	The Percentage of Nitrogen Element	The Percentage of Oxygen Element
CQDs roasted for 5 min	78.03%	0.85%	26.07%
CQDs roasted for 10 min	61.23%	0.44%	38.34%
CQDs roasted for 20 min	79.58%	1.48%	18.94%

## Data Availability

The data used to support the findings of this study can be made available by the corresponding author upon request.

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
