# Peer review of "Carbon Quantum Dots from Roasted Coffee Beans: Their Degree and Mechanism of Cytotoxicity and Their Rapid Removal Using a Pulsed Electric Field"

_foods, 2023, doi:10.3390/foods12122353_

Round 1
Reviewer 1 Report
In this manuscript, the author reports, “Carbon quantum dots from roasted coffee beans: Their degree and mechanism of cytotoxicity and their rapid removal by a pulsed electric field”. The authors should address the following questions before getting a possible publication.
Recommendation: Major revisions are needed as noted.
1. The novelty of the present article should be discussed in the Introduction section.
2. Abstract must be enriched via valuable results which pave the way for understanding the audiences.
3. The rpm of the centrifugation employed should be mentioned in the “The extraction and purification of CQDs” section.
4. The authors are encouraged to represent the elemental composition of the CQDs roasted at different times in a table.
5. The formatting and grammatical errors in the article need to be checked carefully.
6. The author should write the purpose for each test in one/two sentences (in brief) before explaining the results of the characterization techniques.
7. What does the error bar stands for in different Figures throughout the manuscript? It should be mentioned in the Figure captions.
8. The conclusions section needs to improve with selected and highlighted main findings.
9. The authors are encouraged to include (1-2 sentences) future prospects of the present study in the conclusion.
10. The authors have cited relevant references in the Introduction section; however the manuscript needs to be highlighted with recent reports where CQDs were prepared from natural resources further to broaden the impact: The Environmental Science: Nano, 8(4), 848-862; Environmental chemistry letters 18 (2020): 703-727; ACS Applied Polymer Materials, 4(12), 9323-9340; Nanomaterials, 13(3), 554; Biomaterials Advances (2022): 212756; https://doi.org/10.1021/acsabm.2c00664
The above references are for your reference and are not mandatory.
Minor corrections
Author Response
Questions raised by reviewer 1#
In this manuscript, the author reports, “Carbon quantum dots from roasted coffee beans: Their degree and mechanism of cytotoxicity and their rapid removal by a pulsed electric field”. The authors should address the following questions before getting a possible publication.
Recommendation: Major revisions are needed as noted.
1.The novelty of the present article should be discussed in the Introduction section.
Response: Thanks for your kind suggestion.The novelty of the present article was discussed at the last paragraph of Introduction section in our revised manuscript.
2.Abstract must be enriched via valuable results which pave the way for understanding the audiences.
Response: Thank you for raising this question. As suggested, the Abstract was enriched via valuable results which pave the way for understanding the audiences.
3.The rpm of the centrifugation employed should be mentioned in the “The extraction and purification of CQDs” section.
Response: The rpm of the centrifugation employed was mentioned in the “The extraction and purification of CQDs” section in our revised manuscript.
4.The authors are encouraged to represent the elemental composition of the CQDs roasted at different times in a table.
Response: Thank you for these points. Based on your comments, we represented the elemental composition of the CQDs roasted at different times in the table 1.
5.The formatting and grammatical errors in the article need to be checked carefully.
Response: The formatting and grammatical errors in the article were checked carefully in our revised manusript.
6.The author should write the purpose for each test in one/two sentences (in brief) before explaining the results of the characterization techniques.
Response: Thanks, we wrote the purpose for each test in one/tow sentences before explaining the results of the characterization techniques in our revised manuscript.
7.What does the error bar stands for in different Figures throughout the manuscript? It should be mentioned in the Figure captions.
Response: The error bar stands for in different Figures throughout the manuscript was s.d. and we also provided this information in our revised manuscript.
8.The conclusions section needs to improve with selected and highlighted main findings.
Response: Based on your comment, the conclusions section was improved with selected and highlighted main findings in our revised manuscript.
9.The authors are encouraged to include (1-2 sentences) future prospects of the present study in the conclusion.
Response: The future prospects of the present study in the conclusion were provided in our revised manuscript.
10.The authors have cited relevant references in the Introduction section; however the manuscript needs to be highlighted with recent reports where CQDs were prepared from natural resources further to broaden the impact: The Environmental Science: Nano, 8(4), 848-862; Environmental chemistry letters 18 (2020): 703-727; ACS Applied Polymer Materials, 4(12), 9323-9340; Nanomaterials, 13(3), 554; Biomaterials Advances (2022): 212756; https://doi.org/10.1021/acsabm.2c00664. The above references are for your reference and are not mandatory.
Response: Thanks for your suggestion. The recent reports where CQDs were prepared from natural resources were cited in our revised manusript.
Reviewer 2 Report
The paper is interesting and overall, well written. Also, this very interesting investigation is with potential possibility for practical application. The methodology was appropriate where the authors include all type of potential cytotoxic activity and types of cell death.
Minor revision:
NRK cells - specify the chosen cell type …..normal epithelial cells from kidney? and why you used this cells, explain somewhere.
Specify the number and way of cell seeding in the experiments.
In the Materials and Methods section specify which parameters and antibody were used for Western blot analysis.
Author Response
Questions raised by reviewer 2#
The paper is interesting and overall, well written. Also, this very interesting investigation is with potential possibility for practical application. The methodology was appropriate where the authors include all type of potential cytotoxic activity and types of cell death.
Minor revision:
1.NRK cells specify the chosen cell type normal epithelial cells from kidney? and why you used this cells, explain somewhere.
Response: Thanks. Oral administration of CQDs in mice, they mainly accumulated in organs such as the brain, kidneys, and lungs. So NRK cells, normal epithelial cells from kidney, were chosen as the cell model to elucidate the toxicity and toxic mechanism of CQDs. We also provided this information in our revised manusript.
2.Specify the number and way of cell seeding in the experiments.
Response: We specified the number and way of cell seeding in the experiments.
3.In the Materials and Methods section specify which parameters and antibody were used for Western blot analysis.
Response: Thank you for these points. Based on your comments, we specified the parameters and antibodies in our revised manuscript.
Reviewer 3 Report
The manuscript entitled “Carbon quantum dots from roasted coffee beans: Their degree and mechanism of cytotoxicity and their rapid removal by a pulsed electric field” is reviewed, and some major considerations should be addressed by the authors.
1. The title of the study is attractive, but the manuscript fails to define the aim and objective of this work. The authors should describe the neediness and therefore the purpose of this study in the last paragraph of the #Introduction section.
2. How have the authors calculated the size distribution of the CQDs roasted for 5 min (Fig. 1A)? the number of particles are very less in the plot.
3. The FTIR and XRD images should be separately plotted (in bigger size) for clearer representation.
4. What is the reason behind the shift of peaks (5 min, 10 min, and 20 min) of the samples as shown in the #Figure 2D?
5. The authors should prepare the suitable figures (eg., consistency in size, aspect ratios, and labeling) for scientific presentation.
6. How have the authors calculated the rate of apoptosis (%) as shown in the #Fig. 4A? What is the difference in rate of apoptosis and rate of cell death (Figs. 4A&B)? It would be better if the authors provide the cell viability (%) data for better representation.
7. The limitations of this work should be highlighted in the #Conclusion section.
Minor editing of English language required
Author Response
Questions raised by reviewer 3#
The manuscript entitled “Carbon quantum dots from roasted coffee beans: Their degree and mechanism of cytotoxicity and their rapid removal by a pulsed electric field” is reviewed, and some major considerations should be addressed by the authors.
1.The title of the study is attractive, but the manuscript fails to define the aim and objective of this work. The authors should describe the neediness and therefore the purpose of this study in the last paragraph of the #Introduction section.
Response: Thanks for your kind suggestion. As you suggested, we described the neediness and the purpose of this study in the last paragraph of the Introduction section.
2.How have the authors calculated the size distribution of the CQDs roasted for 5 min (Fig. 1A)? the number of particles are very less in the plot.
Response: At least five images of the CQDs roasted for 5min were calculated the size distribution although the number of particles was very less in the plot in each of the images.
3.The FTIR and XRD images should be separately plotted (in bigger size) for clearer representation.
Response: Thanks for your comments. The FTIR and XRD images were separately plotted (in bigger size) for clearer representation in our revised manuscript.
4.What is the reason behind the shift of peaks (5 min, 10 min, and 20 min) of the samples as shown in the #Figure 2D?
Response: The emission peak of carbon dots exhibited the shift of peaks (5 min, 10 min, and 20 min) of the samples as shown in the #Figure 2D. This is a common phenomenon observed in CQDs due to inhomogeneous size distribution or different surface defect states of the particles resulting in differences in electron transition paths. We also provided this sentences in our revised manuscript.
5.The authors should prepare the suitable figures (eg., consistency in size, aspect ratios, and labeling) for scientific presentation.
Response: Based on your comment, we prepared the suitable figures (eg., consistency in size, aspect ratios, and labeling) for scientific presentation in our revised manuscript.
6.How have the authors calculated the rate of apoptosis (%) as shown in the #Fig. 4A? What is the difference in rate of apoptosis and rate of cell death (Figs. 4A&B)? It would be better if the authors provide the cell viability (%) data for better representation.
Response: Phosphatidyl serine is mainly distributed on the inner side of the cell membrane, near the cytoplasm. Apoptosis is one of the ways on cell death. In the early stage of apoptosis, phosphatidylserine is reversed outside the cell membrane, and annexin V can bind to inverted phosphatidylserine. A green fluorescent labeled annexin V-FITC probe, flow cytometry and fluorescence microscopy were used to test whether phosphatidylserine was everted. In the late stage of cell apoptosis, the cell membrane is damaged, and the propidium iodide (PI) probe can penetrate the cell membrane and bind to chromosomes in the nucleus. Necrosis is a passive cell death under environmental stress. Necrosis can also be regulated by fine cellular signaling pathways. Necrosis is caused by the damage of cell membrane, and then the cell permeability changed. PI probe can pass through the cell membrane and combine with the chromosome in the nucleus, so that the cell is labeled by PI. So the rate of cell death may include the rate of apoptosis and necrosis. We also provided the cell viability (%) data in our revised manuscript.
7.The limitations of this work should be highlighted in the #Conclusion section.
Response: Thanks, we highlighted the limitations of this work in the #Conclusion section in our revised manuscript.
Round 2
Reviewer 1 Report
The authors have addressed all the questions raised before therefore the manuscript can be accepted in its present form
Minor
Author Response
Questions raised by reviewer 1#
1.The authors have addressed all the questions raised before therefore the manuscript can be accepted in its present form. Comments on the Quality of English Language (Minor).
Response: Thanks for your kind suggestion. We improved the English Language in our revised manuscript. Several months ago,we also asked a native English speaker to proofread the manuscript and improve the English. We also provided the certificate in cover letter and the attachment.

Reviewer 3 Report
Please correct the the Y-axis legend in Figure 8.
Author Response
Questions raised by reviewer 2#
1.Please correct the the Y-axis legend in Figure 8.
Response: Thanks for your comment. We corrected the Y-axis legend in Figure 8.